# Identification of Causal Relationships between Gut Microbiota and Influenza a Virus Infection in Chinese by Mendelian Randomization

**DOI:** 10.3390/microorganisms12061170

**Published:** 2024-06-08

**Authors:** Qijun Liao, Fuxiang Wang, Wudi Zhou, Guancheng Liao, Haoyang Zhang, Yuelong Shu, Yongkun Chen

**Affiliations:** 1School of Public Health (Shenzhen), Shenzhen Campus of Sun Yat-sen University, Shenzhen 518107, China; liaoqj@mail2.sysu.edu.cn (Q.L.); wangfx6@mail2.sysu.edu.cn (F.W.); zhouwd5@mail2.sysu.edu.cn (W.Z.); liaogch5@mail2.sysu.edu.cn (G.L.); 2BGI Genomics, Shenzhen 518085, China; 3School of Data and Computer Science, Sun Yat-sen University, Guangzhou 510006, China; zhanghaoyang0@hotmail.com; 4Key Laboratory of Pathogen Infection Prevention and Control (MOE), State Key Laboratory of Respiratory Health and Multimorbidity, National Institute of Pathogen Biology, Chinese Academy of Medical Sciences and Peking Union Medical College, Beijing 102629, China; 5Guangdong Provincial Key Laboratory of Infection Immunity and Inflammation, Department of Pathogen Biology, School of Basic Medical Sciences, Shenzhen University Medical School, Shenzhen University, Shenzhen 518055, China

**Keywords:** H7N9, H1N1, influenza A virus, Mendelian randomization, microbiota

## Abstract

Numerous studies have reported a correlation between gut microbiota and influenza A virus (IAV) infection and disease severity. However, the causal relationship between these factors remains inadequately explored. This investigation aimed to assess the influence of gut microbiota on susceptibility to human infection with H7N9 avian IAV and the severity of influenza A (H1N1)pdm09 infection. A two-sample Mendelian randomization analysis was conducted, integrating our in-house genome-wide association study (GWAS) on H7N9 susceptibility and H1N1pdm09 severity with a metagenomics GWAS dataset from a Chinese population. Twelve and fifteen gut microbiotas were causally associated with H7N9 susceptibility or H1N1pdm09 severity, separately. Notably, *Clostridium hylemonae* and *Faecalibacterium prausnitzii* were negative associated with H7N9 susceptibility and H1N1pdm09 severity, respectively. Moreover, *Streptococcus peroris* and *Streptococcus sanguinis* were associated with H7N9 susceptibility, while *Streptococcus parasanguini* and *Streptococcus suis* were correlated with H1N1pdm09 severity. These results provide novel insights into the interplay between gut microbiota and IAV pathogenesis as well as new clues for mechanism research regarding therapeutic interventions or IAV infections. Future studies should concentrate on clarifying the regulatory mechanisms of gut microbiota and developing efficacious approaches to reduce the incidence of IAV infections, which could improve strategy for preventing and treating IAV infection worldwide.

## 1. Introduction

Influenza A viruses (IAVs) are primarily transmitted among aquatic birds and poultry but can sporadically cross species barriers to infect humans. Certain subtypes, such as H1N1 and H3N2, are capable of adapting to humans, enabling person-to-person transmission and triggering pandemics. These strains may evolve into seasonal influenza viruses [1]. However, avian IAVs, such as H5N1 and H7N9, fail to fully adapt to humans, lacking efficient human-to-human transmission after spillover infection. Despite this, spillover events may result in a high case fatality rate and harbor the potential for a new pandemic if the viruses continually evolve and adapt to humans [2]. According to the World Health Organization, seasonal IAV accounts for an estimated one billion infections, 3 to 5 million severe cases, and approximately 290,000 to 650,000 deaths each year [3]. Notably, avian IAV H7N9 triggered five outbreaks in China from 2013 to 2017, resulting in 1568 confirmed infections with a fatality rate of approximately 39%. To alleviate the public health impact of IAVs, it is imperative to investigate host factors influencing susceptibility to avian IAVs and the severity of seasonal IAVs. Recent research has underscored the role of innate immune system proteins (e.g., MxA and BTN3A3) in reducing human susceptibility to avian IAV H7N9 [4,5], while genetic variations in host genes, such as *IFITM3* and *IRF7,* predispose individuals to severe disease caused by H1N1pdm09 [6,7]. This emphasizes the critical role of host factors in impeding the spread of IAV strains and underscores the potential of the human immune system to defend against such infections.

The gut microbiota, a complex and highly diverse microbial community, serves as an essential mediator linking diseases to human genome evolution and is closely associated with the development and progression of many diseases [8,9]. Numerous studies have recognized the broad-reaching immune impact of the gut microbiota on pulmonary health [10,11,12]. A systematic review of gut microbiota changes in respiratory tract infection (RTI) patients consistently revealed decreased diversity, with depletion of *Firmicutes*, *Lachnospiraceae*, *Ruminococcaceae*, and *Ruminococcus*, and enrichment of *Enterococcus* [13]. Fecal transfer experiments have indicated that gut microbes from mice surviving H7N9 infection can confer resistance to naive recipient mice challenged with IAV [14]. Additionally, patients with IAV infection may experience symptoms resembling gastroenteritis [15], indicating the importance of the gut–lung axis in maintaining lung immunity homeostasis during influenza infection. Investigations have revealed significant changes in the composition of intestinal microbiota in individuals affected by different IAV subtypes, including H7N9 and H1N1 [16,17,18]. However, these studies have only established associations between gut microbiota and various IAV subtypes. Due to the potential confounding factors and reverse causality, the causal relationship between gut flora and IAV infection remains unclear.

Mendelian randomization (MR) provides an efficient approach to evaluating causal effects using genetic variants as instrumental variables (IVs) [19]. Genetic variants are randomly assigned during meiosis and not influenced by traditional confounding factors, such as environment, socioeconomic status, and behavior. Furthermore, genetic variants remain stable after birth, enabling the evaluation of chronologically plausible associations. Therefore, MR can overcome confounding and reverse causality issues inherent in traditional observational studies. MR analysis has been widely applied to assess causal effects between gut microbiota and diseases, such as autoimmune diseases and coronavirus disease 2019 (COVID-19) [20,21]. Xu et al. employed MR to investigate the causal effect of gut microbiota on seasonal influenza and influenza-induced pneumonia in Finland. In their study, the identification of intestinal flora was limited to the genus level [22]. In this study, we performed a systematic two-sample MR study to explore the causal effect of species-level gut microbiota on avian IAV H7N9 susceptibility and H1N1pdm09 severity in Chinese. Note that genome-wide association studies (GWAS) of H7N9 susceptibility have been restricted to Chinese populations thus far. The outcomes of this study might offer new insights for personalized IAV treatment through the regulation of gut microbiota.

## 2. Materials and Methods

### 2.1. Study Design

First, we conducted a two-sample MR analysis to investigate potential causal relationships between gut microbial features and IAV infection. Three different MR methods (refer to Methods further below) were performed to increase the robustness of the results and avoid bias. The selection of IVs adhered to three critical assumptions [19]: (i) Relevance: the IVs were associated with the exposure of interest; (ii) Independence: the IVs were independent of other confounding factors that affect both the exposure and outcome; (iii) Exclusion restriction: the IVs were required to influence the outcome solely through the studied exposure. The stringent criteria for IV selection were essential to ensure the validity and robustness of the MR analysis results. Subsequently, protein–protein interaction (PPI) and functional enrichment analyses were carried out to investigate possible biological connections between gut microbiota and influenza infection. The study design is illustrated in Figure 1.

### 2.2. Gut Microbiome Data

The GWAS summary data for human gut microbiome composition were obtained from an extensive study [23], involving 1539 adult Chinese individuals with both blood and fecal samples available. Each participant underwent high-depth whole-genome and whole-metagenomic sequencing with a mean sequencing depth of 42× for the whole genome, ensuring precise genotyping. This dataset was composed of 500 microbial features, including 401 taxa (248 species, 95 genera, 32 families, 14 orders, 3 classes, and 9 phyla) and 99 module functions (MFs), all integrated into the analysis. The GWAS results for microbial features were meticulously adjusted for various factors, such as age, gender, body mass index, defecation frequency, stool form, 12 dietary and lifestyle factors, and the top four principal components (PCs), to minimize the effects of potential confounding factors [23].

### 2.3. H7N9 Susceptibility and H1N1 Severity GWAS Data

The summary statistics for H7N9 susceptibility were retrieved from a previous GWAS involving 217 H7N9 patients and 116 healthy poultry workers, all of Chinese descent [5]. Real-time reverse transcription polymerase chain reaction (RT–PCR) was used for diagnosing H7N9 infection, following the Diagnostic and Treatment Protocol for Human Infection with Avian Influenza A (H7N9) (http://www.nhc.gov.cn/gjhzs/s7952/201304/34750c7e6930463aac789b3e2156632f.shtml accessed on 30 May 2018). Similarly, summary statistics for H1N1 severity were collected from another preceding GWAS involving 165 Chinese H1N1pdm09 patients, confirmed by positive results of real-time RT–PCR [24]. Patients with mild symptoms (*n* = 95) were defined as outpatients not admitted to hospitals, while those with severe symptoms (*n* = 70) were hospitalized patients meeting specific criteria outlined in the Protocol for diagnosis and treatment of influenza (2019 version) [25]. All participants underwent high-depth whole-genome sequencing, with an average sequencing depth exceeding 30×, to detect potential variants and for genotyping. Subsequently, GWAS summary statistics were generated using logistic regression analysis, adjusted for gender, age, and PCs.

### 2.4. Selection of IVs

The single nucleotide polymorphisms (SNPs) associated with microbiome features were chosen as eligible IVs according to several criteria. Firstly, SNPs with a *p* value < 1 × 10^−5^ were chosen to represent a broad range of microbiome [23,26]. To ensure independence, linkage disequilibrium (LD) clumping was performed (*r*^2^ < 0.01 within a 1000 kb window) using the 1000 Genomes EAS population as a reference. SNPs with palindromic A/T or G/C alleles were excluded to prevent strand orientation or allele coding issues. SNPs associated with the outcome (*p* < 1 × 10^−4^) were also excluded. The strength of IVs was assessed using *F*-statistics, calculated using the formula [27]: *F* = *R*^2^(*N* − 1 − *k*)/(1 − *R*^2^)*k*(1)
where *N* represents the sample size, *k* indicates the number of IVs, and *R*^2^ stands for the proportion of variance explained. For a SNP, *R*^2^ was calculated using the equation:*R*^2^ = 2 × *EAF* × (1 − *EAF*) × *β*^2^/[2 × *EAF* × (1 − *EAF*) × *β*^2^ + 2 × *EAF* × (1 − *EAF*) × *N* × *SE*^2^](2)
where *EAF* is the effect allele frequency, *β* is the effect size, and *SE* is the standard error of effect size. SNPs with *F*-statistics < 10 were deemed insufficient in strength and were subsequently removed [28]. The independence of selected SNPs allowed for the calculation of the combined *R*^2^ as the sum of individual *R*^2^ values under the assumption of an additive model.

### 2.5. Two-Sample MR Analysis

The inverse variance weighted (IVW) test served as the primary method to evaluate causal effects, aggregating Wald ratios of IVs in a fixed-effect meta-analysis model, providing reliable results under the assumption of no horizontal pleiotropy for each IV [29]. Additionally, MR–Egger regression and the weighted median (WM) method were used for supplementary analysis. Suppose there are *M* genetic variants, where *j* represents the *j*th variant, with *x* denoting the exposure, and *y* signifying the outcome. The IVW method is a basic model assuming that pleiotropy does not exist or is zero. It estimated the causal effect ($\beta_{xy\;}$) by integrating the effect ratio ($\theta_{j}=\beta_{yj\;}/\beta_{xj\;}$) of genetic variant on outcome ($\beta_{yj\;}$) and exposure ($\beta_{xj\;}$) using inverse variance weighting. IVW estimates can also be obtained using the following weighted linear regression model without intercept terms:(3)$${\hat{\beta }}_{yj}=b{\hat{\beta }}_{xj}+\varepsilon_{j},\;\varepsilon_{j}\;\sim \;N\left(0,\;{se\left(\beta_{yj\;}\right)}^{2}\right)$$

The estimate of the slope parameter *b* is the causal effect, and the weight is the inverse of the variance of the genetic association effect corresponding to the outcome. In order to control the uncorrelated pleiotropy, MR–Egger identifies the violations of pleiotropy and heterogeneity by incorporating an intercept in the IVW model [30]. The regression model is as follows:(4)$${\hat{\beta }}_{yj}=b_{0}+b{\hat{\beta }}_{xj}+\varepsilon_{j},\;\varepsilon_{j}\;\sim \;N\left(0,\;{se\left(\beta_{yj\;}\right)}^{2}\right)$$

Because some IVs may be pleiotropic, the WM method estimates the causal effect from the weighted median of the effect ratio ($\theta_{j}$). The WM method offers consistent estimates even when up to 50% of the IVs are invalid [31]. Following the application of the Bonferroni correction, we established a statistically significant threshold of *p* = 1 × 10^−4^ (0.05/500). The associations with *p* < 0.05 but above 1 × 10^−4^ were considered nominally significant.

### 2.6. Sensitivity Analysis

Sensitivity analyses were conducted to assess the robustness of the MR analysis. MR–Egger intercept tests and MR–Pleiotropy Residual Sum and Outlier (MR–PRESSO) global tests were used to examine horizontal pleiotropy [32,33]. Additionally, Cochran’s Q statistic and a funnel plot were used to examine heterogeneity [34]. A leave-one-out analysis was conducted to demonstrate that inferred causal relationships were not influenced by a single SNP [32]. All MR analyses were carried out using R software (version 4.1.2) and the “TwoSampleMR” package (version 0.5.6).

### 2.7. Biological Annotation

ANNOVAR (version 20180416) was used to functionally annotate variants with *p* < 1 × 10^−4^ in H7N9 susceptibility and H1N1 severity GWAS, and variants with *p* < 1 × 10^−5^ in the gut microbiome GWAS [35]. PPI networks were constructed using shared positional mapped genes between IAVs and the gut microbiome. These networks were predicted using the Search Tool for the Retrieval of Interacting Genes (STRING, version 12.0) online database [36]. A combined score ≥ 0.4 was chosen for construction of the PPI networks. We also extracted the functional enrichment results for human phenotype (Experimental Factor Ontology [37] and Human Phenotype Ontology [38]) from STRING, which could provide insight into the associations of genes with various phenotypes. The connected PPI networks were visualized using Cytoscape software (version 3.8.0) [39]. The top 10 nodes were identified as hub genes using the Maximal Clique Centrality (MCC) method with the CytoHubba plug-in (version 0.1) [40]. The effect of hub genes on multiple phenotypes was evaluated through Phenome-Wide Association Study (PheWAS) by examining the pleiotropy of hub genes in the summary statistics for 4756 complex traits and diseases across 28 domains using the GWAS ATLAS [41]. PheWAS offered a more comprehensive understanding of the biological significance of these genes and helped elucidate the MR results. The statistically significant threshold was defined as 1.05 × 10^−5^ (0.05/4756). IVs for pairs of microbial features and IAVs with significant causality were annotated using ANNOVAR. Additionally, Gene Ontology (GO) [42,43] and Kyoto Encyclopedia of Genes and Genomes (KEGG) [44] pathway enrichment analyses were performed for the positional mapped genes using DAVID (https://david.ncifcrf.gov/tools.jsp accessed on 21 January 2024) [45,46] to reveal the potential underlying biological pathways or functions of causal associations. GO is utilized as a bioinformatics tool for annotating genes and analyzing the biological processes they are involved in. It classifies gene functions into biological processes (BP), molecular functions (MF), and cellular components (CC). KEGG is a database for analyzing molecular signaling pathways and interactions in biological systems. A significance threshold of *p* < 0.05 and Fold Enrichment > 2.5 was considered statistically significant. The enrichment analysis results are visualized by SRplot web server (http://www.bioinformatics.com.cn/SRplot accessed on 21 January 2024) [47].

## 3. Results

### 3.1. Effect of Gut Microbiota on H7N9 Susceptibility

The results of the MR analysis concerning H7N9 susceptibility are detailed in Appendix A. The genetic IVs varied from 2 to 29 across different gut microbiome features. *F*-statistics for the human gut microbiota ranged from 20.17 to 37.02, all surpassing 10, indicating a reduced susceptibility to weak instrument bias. Initially, the IVW method identified 14 gut bacterial taxa associated with H7N9 susceptibility (Figure 2, Appendix A). The effect directions estimated by MW were consistent with those derived from IVW. However, the MR–Egger method revealed inconsistent directions for two gut bacterial taxa, including the genus *Rahnella* and *Coprococcus catus*. Subsequently, 12 gut bacterial taxa were found to be nominally causally associated with H7N9 susceptibility with statistical significance. It is worth noting that *Clostridium hylemonae* (*β* = −0.335, 95% CI: −0.622 to −0.049, *p* = 0.022) and *Clostridium ramosum* (*β* = −0.390, 95% CI: −0.684 to −0.096, *p* = 0.009), known for synthesizing short-chain fatty acids (SCFAs), exhibited a negative correlation with the susceptibility to H7N9. Additionally, *Streptococcus peroris* (*β* = −0.327, 95% CI: −0.648 to −0.006, *p* = 0.046) and *Streptococcus sanguinis* (*β* = −0.552, 95% CI: −1.017 to −0.087, *p* = 0.020)) were associated with a reduced risk of H7N9 susceptibility, while *Streptococcus mitis* (*β* = 0.504, 95% CI: 0.022 to 0.986, *p* = 0.041) showed an increased risk of H7N9 susceptibility. Scatter plots are displayed in Appendix A.

In the sensitivity analysis (Table 1), both the MR–PRESSO Global test and MR–Egger intercept test indicated a limited impact of horizontal pleiotropy. Furthermore, Cochran’s Q test suggested no significant heterogeneity. The leave-one-out analysis indicated that no SNPs significantly influenced the overall result (Appendix A). Most funnel plots demonstrated symmetry (Appendix A), reinforcing the robustness and reliability of the MR analysis results.

### 3.2. Effect of Gut Microbiota on H1N1 Severity

Causal effects of all gut microbiota on H1N1 severity are presented in Appendix A. The genetic IVs for each gut microbiome feature ranged from 7 to 31, with *F*-statistics ranging from 23.38 to 30.70, surpassing the empirical threshold of 10. The IVW analysis identified 18 gut bacterial taxa associated with H1N1 severity. However, alternative MR analysis methods revealed inconsistent effect directions for three bacterial species: *Roseburia intestinalis*, *Treponema vincentii*, and *Veillonella atypica*. Ultimately, 15 gut bacterial taxa met the criteria as significant contributors to the development of H1N1 severity (Figure 3, Appendix A). In particular, *Faecalibacterium prausnitzii* (*β* = −0.394, 95% CI: −0.774 to −0.013, *p* = 0.043), playing a key role in the biosynthesis of SCFAs, exhibited a negative association with the risk of H1N1 severity. Moreover, *Streptococcus parasanguinis* (*β* = −0.574, 95% CI: −1.004 to −0.144, *p* = 0.009) and *Streptococcus suis* (*β* = −0.455, 95% CI: −0.865 to −0.044, *p* = 0.030) were also correlated with a reduced risk of H1N1 severity. Scatter plots are available in Appendix A.

Based on the results of sensitivity analysis, no significant horizontal pleiotropy or heterogeneity was detected using the MR–Egger intercept test, MR–PRESSO global test and Cochran’s Q test (Table 2). Furthermore, the leave-one-out analysis and funnel plots demonstrated the stability of the findings (Appendix A).

### 3.3. Biological Annotation

We identified a total of 87 shared GWAS significant genes connecting H7N9 susceptibility and microbiota features (Appendix A) and 30 shared genes of H1N1 severity and microbiota features (Appendix A). The PPI network analysis of both gene sets using the STRING database revealed a significantly higher number of interactions than expected by chance. By evaluating the enrichment of observed edges compared to expected edges, we obtained a PPI *p*-value of 8.43 × 10^−8^ for 87 proteins (Appendix A) and 3.70 × 10^−3^ for 30 proteins (Appendix A), indicating a biological connection to some extent. According to STRING, previous studies found that these genes were associated with several phenotypes at a false discovery rate (FDR) < 0.05. For the overlapping proteins between H7N9 susceptibility and microbiota features, we identified a significant enrichment of 44 phenotypes (Appendix A), including gut microbiome measurement, susceptibility to infectious disease measurement, respiratory disease biomarker, etc. In contrast, we observed a significant enrichment of only three phenotypes among the overlapping proteins between H1N1 severity and microbiota features (Appendix A).

The connected PPI networks were visualized using Cytoscape (Figure 4), and hub genes were screened (Appendix A). Except for *ZBTB18* and *C1orf100*, all hub genes were associated with multiple phenotypes (*p* < 1.05 × 10^−5^). In the PPI network linking H7N9 susceptibility to microbiota features, PheWAS results revealed that 8 out of 10 hub genes (*LRP1B*, *ROBO2*, *USH2A*, *FOXP1*, *TENM4*, *CACNA1C*, *PCLO*, and *AGRN*) showed enrichment with genetic signals associated with the metabolic or nutritional domain. Moreover, 4 of the 10 hub genes (*LRP1B*, *FOXP1*, *CACNA1C*, and *KALRN*) were enriched with genetic signals associated with the immunological domain (Appendix A). Similarly, 8 of 10 hub genes (*PTPRD*, *RARB*, *PPARGC1A*, *SORCS2*, *MED15*, *DLGAP1*, *PLCB4*, and *DGKB*) in the network connecting H1N1 severity to microbiota features showed enrichment with the metabolic or nutritional domain. However, only one hub gene (*PPARGC1A*) was enriched with the immunological domain (Appendix A). In summary, PPI networks and PheWAS analysis uncover the connection between IAV infection and microbial features, which are associated with hub genes, as well as the metabolic or nutritional domain and the immunological domain.

A total of 170 IVs for the 12 pairs of microbial features and H7N9 susceptibility with potential causality were mapped onto 216 genes (Appendix A), and 232 IVs for the 15 pairs of microbial features and H1N1 severity were mapped onto 304 genes (Appendix A). The GO analysis showed that both gene sets were significantly enriched in several terms, including cell–cell junction assembly, adherens junction organization, cell–cell adhesion mediated by cadherin, cell–cell adhesion via plasma-membrane adhesion molecules, calcium-dependent cell–cell adhesion via plasma membrane cell adhesion molecules, and homophilic cell adhesion via plasma membrane adhesion molecules. Additionally, the 216 mapped genes were significantly enriched in galactosylceramide catabolic process, phosphorylation, while the 304 mapped genes were significantly enriched in transmembrane receptor protein tyrosine kinase signaling pathway, Ras protein signal transduction, positive regulation of T-helper cell differentiation, etc. (Figure 5, Appendix A). In the KEGG analysis, the 216 mapped genes did not show significantly enriched pathways, whereas the 317 mapped genes were significantly enriched in the Rap1, Apelin, and calcium signaling pathways (Appendix A).

## 4. Discussion

Our research revealed that 12 specific bacterial features were causally associated with H7N9 susceptibility, while 15 bacterial features were causally associated with the severity of H1N1, underscoring the significance of the gut–lung axis in regulating immune responses during IAV infections. We also provided a possible biological interpretation of the association between gut microbiota and IAV infection using gene- and gene-set-based analyses on multi-omics data. 

The complex and diverse microbial ecosystem in the gastrointestinal tract plays a pivotal role in maintaining human health. Recent studies have demonstrated the intricate interplay between gut microbiota and immune system regulation, extending its impact beyond the gut to encompass the entire body, including distant organs such as the lungs [48,49,50]. The gut–lung axis, representing the interaction between gastrointestinal bacteria and the pulmonary system, can modulate inflammatory activity at both local and systemic levels [51]. Notably, in the treatment of IAV-induced viral respiratory infections, the significant disruption of the gut microbial ecosystem by antibiotics can compromise the innate and adaptive defenses of the host, emphasizing the essential role of the gut–lung axis in IAV infection [52,53]. Furthermore, emerging evidence has underscored the critical role of microbiota composition in regulating the development of virus-specific CD4 and CD8 T cells, as well as antibody responses, in respiratory influenza virus infections [53]. Disruptions in gut microbes could potentially impair immune cell migration to the lungs, thereby increasing susceptibility to respiratory tract infections [54,55]. This observation emphasized the pivotal role of gut microbiota in shaping host immune responses and underscored the potential for modulating microbiota composition as a new therapeutic approach for treating influenza and other respiratory viral infections.

There is mounting evidence linking the composition of gut microbiota to IAV infections, with the fecal microbiomes of H1N1 and H7N9 patients showing a noticeable decrease in bacterial diversity [16,17]. It is worth noting that gut dysbiosis, often characterized by reduced microbial diversity, is commonly associated with inflammatory and autoimmune disorders [51]. In patients with laboratory-confirmed H1N1, a reduction in the phyla *Actinobacteria* and *Firmicutes*, along with the genera *Faecalibacterium* and *Streptococcus*, was observed compared to healthy controls [16]. Similarly, in a study involving 26 patients infected with H7N9, a decrease in the phyla of *Bacteroidetes* and the genera of *Eubacterium*, *Roseburia*, and *Faecalibacterium*, as well as the species of *Roseburia intestinalis* and *Faecalibacterium prausnitzii*, was noted, while an enrichment was observed for the genus of *Veillonella* [17].

Consistent with previous research, our study identified that six out of eight bacterial taxa negatively associated with H1N1 severity belonged to the phylum *Firmicutes*, with the exception of *Bacteroides thetaiotaomicron* and *Bacteroides xylanisolvens*. Similarly, apart from *Citrobacter youngae*, the other four bacterial taxa negatively linked to H7N9 susceptibility also belonged to *Firmicutes*. Previous studies have reported a reduction in *Firmicutes* among patients with RTIs, including H1N1 infection, COVID-19, tuberculosis (TB), community-acquired pneumonia, and recurrent respiratory tract infections [13]. Certain gut microbes, notably *Faecalibacterium prausnitzii* and *Clostridium hylemonae* of the *Firmicutes* phylum, possess a strong ability to produce SCFAs, such as acetate, propionate, and butyrate, through the fermentation of dietary fiber and resistant starch [56,57]. Microbiota-derived butyrate activates the nuclear receptor peroxisome proliferator activated receptor γ in colonic epithelial cells, shifting their energy metabolism towards fatty acid oxidation and oxidative phosphorylation in mitochondria, which consumes high levels of oxygen [58,59,60]. This suggests that SCFAs promote gut homeostasis through a positive feedback loop by limiting the luminal bioavailability of oxygen. SCFAs can also signal through the membrane receptor G protein-coupled receptor (GPR) to activate signaling pathways that regulate immune functions [61]. Research has shown that acetate could protect against respiratory syncytial virus-induced disease by activating GPR43 and modulating type 1 responses in lung epithelial cells [62]. Moreover, SCFAs can induce both effector and regulatory T cells to regulate the immune system by inhibiting the direct histone deacetylase (HDAC) in T cells and enhancing the mTOR-S6K pathway [63]. In our study, *Faecalibacterium prausnitzii*, a representative bacterium of the phylum *Firmicutes*, was also found to be negatively correlated with H1N1 severity. *Faecalibacterium prausnitzii* is considered as an anti-inflammatory probiotic that provides defense against various gastrointestinal disorders [64]. Reduced levels of *Faecalibacterium prausnitzii* have been observed in patients with COVID-19, TB, asthma, and cystic fibrosis [65,66,67,68]. Additionally, a study investigating the correlations between gut microbiota and persistent symptoms in recovered COVID-19 patients revealed a negative correlation between chest tightness after activity and the relative abundance of *Faecalibacterium prausnitzii* [69].

Our results indicated that the species *Streptococcus peroris* and *Streptococcus sanguinis*, as well as *Streptococcus parasanguinis* and *Streptococcus suis*, were negatively correlated with H7N9 susceptibility and H1N1 severity, respectively. However, concerning the genus *Streptococcus*, different outcomes were observed in individuals infected with H1N1 and H7N9 compared to the control group, showing decreased levels for H1N1 and elevated levels for H7N9 [16,17]. Animal studies have indicated that pre-exposure to *Streptococcus suis* improved survival in mice co-infected with the influenza virus, with the upregulated innate immunity potentially playing a significant role in reducing mortality when the bacteria were administered before viral infection [70]. Furthermore, a clinical trial demonstrated that the estimated risk of respiratory failure during the course of COVID-19 was significantly lower by eightfold in the group receiving oral bacteriotherapy with streptococcal-containing preparations compared to the untreated group. Conversely, the incidence of ICU admission and mortality was higher among patients not treated with oral bacteriotherapy [71]. The bacterial strains present in the bacterial formulation enhanced the production of both nuclear factor erythroid 2p45-related factor 2 and its target Heme oxygenase-1, exerting antiviral effects through the reduction of oxidative stress [72]. Interestingly, Xu et al. reported that the genus *Streptococcus* was negatively associated with influenza outcomes in FinnGen cohorts, consistent with our findings in the Chinese population [22]. However, we also found that *Streptococcus mitis* was positively associated with H7N9 susceptibility. *Streptococcus mitis* is a predominant cause of infective endocarditis and bacteremia. It is notable that the majority of virulence factors identified in the *Streptococcus pneumoniae* genome are shared with *Streptococcus mitis* [73]. These findings suggest that not all *Streptococcus* species provide protection against influenza. In particular, *Streptococcus mitis* appears to be a potential risk factor for susceptibility to H7N9.

In our study, we identified hub genes such as *LRP1B* and *PTPRD* that serve as key connections between IAV infection and microbiota features. *LRP1B*, which belongs to the low-density lipoprotein (LDL) receptor family, plays diverse roles in normal cell function and development by interacting with multiple ligands. The LDL receptor family has roles related to the clearance of extracellular ligands and is proposed to be involved in extracellular signal transduction [74]. A previous GWAS study demonstrated an association between *LRP1B* and Influenza A (H1N1) infection [75]. Depletion of *LRP1B* has been shown to reduce IAV A/Puerto Rico/8/34 H1N1 infection [76] and influenza A/WSN/33 replication [77]. *PTPRD* encoded a member of the protein tyrosine phosphatase (PTP) family. PTPs are known to be signaling molecules that regulate a variety of cellular processes, including cell growth, differentiation, mitotic cycle, and oncogenic transformation [78]. *PTPRD* is involved in various signaling pathways, including PTPRD/STAT3/JAK and PTPRD/PD-1/PD-L1 axes [79,80,81,82]. These pathways are crucial in regulating IAV replication and anti-IAV immunity [83,84]. Moreover, PheWAS analysis revealed connections between hub genes and metabolic, nutritional, and immunological traits. GO enrichment highlighted a significant association between genes derived from IAVs and the BP of adherens junctions. Adherens junctions are formed by the transmembrane adhesion molecule vascular endothelial (VE)-cadherin and its cytoplasmic tail binding molecules, like β-catenin and plakoglobin, which anchor to actin via α-catenin and stabilize the junction [85]. The Rap1 signaling pathway, identified through KEGG enrichment, plays a critical role in regulating cell adhesion and cell–cell junction formation by modulating the function of integrins and other adhesion molecules in various cell types. Notably, gut microbiota could influence endothelial cell function at distant sites, such as the liver [86]. During IAV infection, the virus affects VE cells by inducing β-catenin degradation in adherens junctions, which is one of the key mechanisms leading to vascular hyperpermeability in severe influenza [87]. Therefore, a plausible mechanism for the causal relationship between gut microbiota and IAV infection may involve the regulation of adherens junctions in endothelial cells during the course of IAV infection.

This study had several strengths. Firstly, we utilized the largest-scale metagenomics GWAS dataset available for individuals of Chinese ancestry. Despite the relatively small sample size of the H1N1 and H7N9 GWAS, they represented the most extensive GWAS study on avian IAV susceptibility conducted to date. Secondly, the application of metagenomic sequencing techniques allowed for precise bacterial classification down to the species level. Through the analysis of biologically relevant samples, metagenomes provided a comprehensive view of the gut microbial community, offering unparalleled insights into bacterial diversity and composition. Thirdly, our study adopted a two-sample MR approach to estimate causal effects and integrated three sensitivity analysis techniques to ensure robustness and mitigate potential pleiotropy from the IVs. By employing a rigorous analytical framework, the study aimed to provide accurate quantitative estimates of the causal relationship, thereby enhancing confidence in the validity of the findings. Our results revealed the correlations between gut microbiota composition and avian IAV infection, suggesting the potential utility of gut microbiota as a targeted risk assessment tool for the disease and as a potential therapeutic strategy.

There were several limitations in this study. Firstly, the ethnic homogeneity of the research population necessitates caution when extrapolating our findings to individuals of different races. Secondly, although we adopted one of the largest gut metagenomics and IAV GWAS datasets to date, the sample size remained modest, and the number of loci examined was relatively limited. Therefore, further research based on larger GWAS datasets is crucial to validate our observations and establish the generalizability of the findings across diverse populations. Thirdly, our findings were not corroborated by in vivo flora colonization assay.

Despite these limitations, we believe that our findings provided new clues for further investigation of microbial function via microbiota colonization in vivo. Focusing on gut microbiota could represent an innovative approach to the prophylaxis and therapy of IAV infections.

## 5. Conclusions

Our MR study has successfully identified the potential causal effects of 12 and 15 gut microbial features on H7N9 infection and H1N1pdm09 severity, respectively. In particular, *Clostridium hylemonae* and *Faecalibacterium prausnitzii*, which promote the production of SCFAs, were negatively associated with H7N9 susceptibility and H1N1pdm09 severity separately. These findings not only highlight the significant implications for understanding the pathogenesis of IAV infection but also provide valuable clues for future research to elucidate the role of gut microbiota in infectious diseases.

In summary, our study not only expands our knowledge concerning the impact of the gut microbiota on H7N9 susceptibility and H1N1pdm09 severity but also underscores the potential for targeting gut microbial composition based on these findings. The causally related bacterial groups open a promising avenue for the development of novel preventative and therapeutic strategies against IAV and other respiratory tract infections. An in-depth comprehension of the foundational mechanisms will facilitate the development of efficacious strategies to modulate the gut microbiome, thereby reducing the incidence and severity of IAV infections.

## Figures and Tables

**Figure 1 microorganisms-12-01170-f001:**
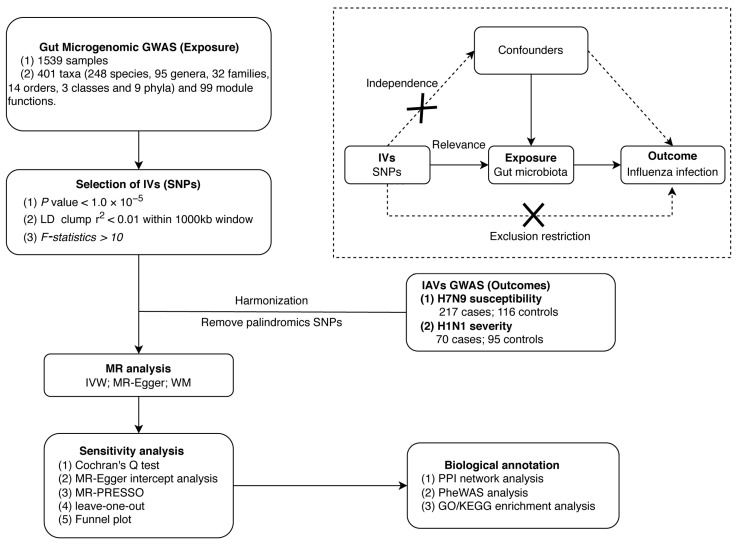
Overview of the process of Mendelian randomization analysis and major assumptions.

**Figure 2 microorganisms-12-01170-f002:**
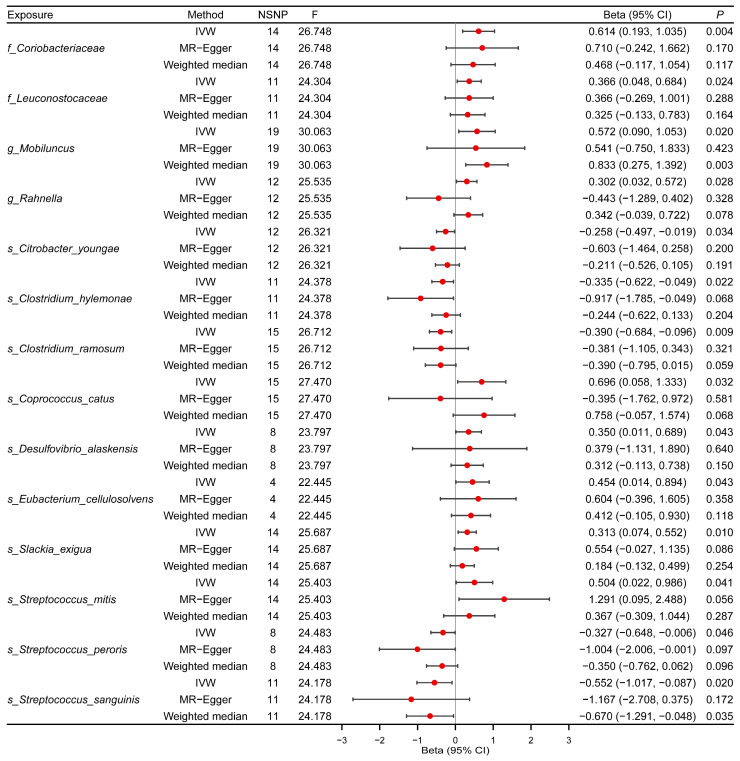
Causal effects of gut microbiota on H7N9 susceptibility. Summary of Mendelian randomization (MR) estimates derived from inverse variance weighted (IVW), weighted median (WM), and MR–Egger analyses. CI denotes confidence interval; OR, odds ratio; SNPs, single nucleotide polymorphisms. “s_”, “g_”, “f_” are species, genus, and family, respectively.

**Figure 3 microorganisms-12-01170-f003:**
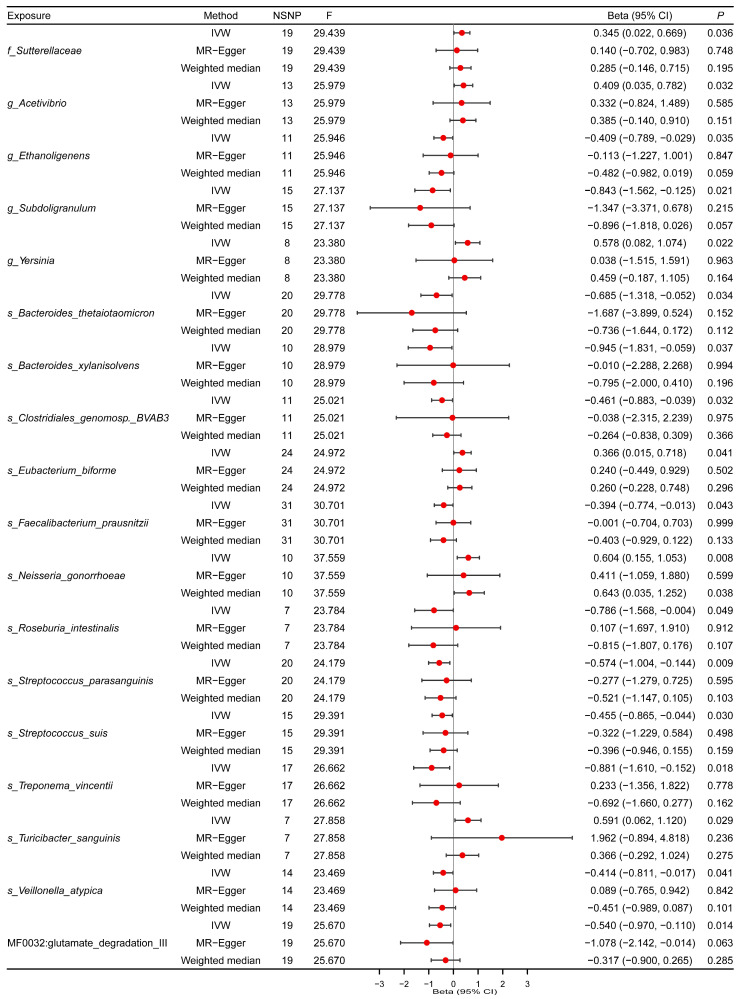
Causal effects of gut microbiota on H1N1 severity. Summary of Mendelian randomization (MR) estimates obtained from inverse variance weighted (IVW), weighted median (WM), and MR–Egger analyses. CI, confidence interval; SNPs, single nucleotide polymorphisms. “s_”, “g_”, “f_” are species, genus, and family, respectively.

**Figure 4 microorganisms-12-01170-f004:**
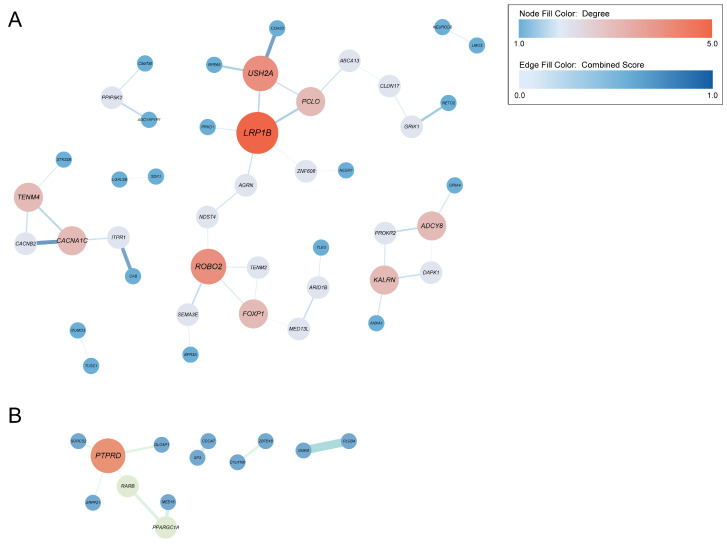
Visualization of connected protein-protein interaction networks using Cytoscape. (**A**) The network of overlapped genes between H7N9 susceptibility and microbiota features. (**B**) The network of overlapped genes between H1N1 severity and microbiota features. Node size and color correspond to the respective degrees, while edge weight and color are proportional to the STRING confidence score.

**Figure 5 microorganisms-12-01170-f005:**
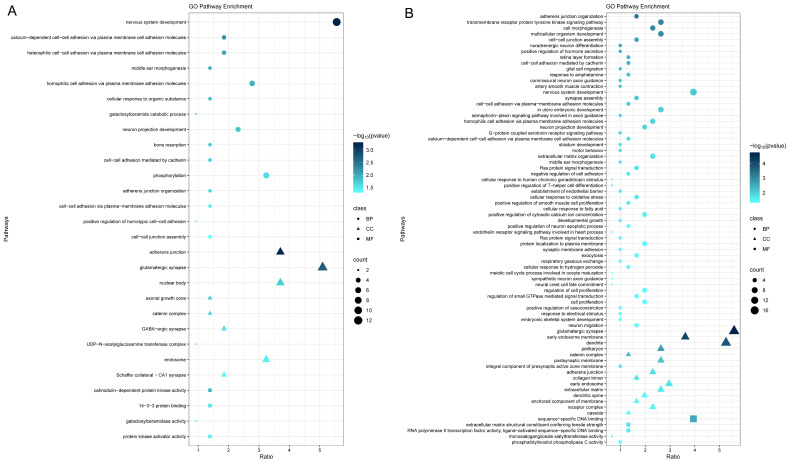
Gene Ontology pathway enrichment analysis performed on (**A**) 216 genes annotated from instrumental variables for pairs of microbial features and H7N9 susceptibility with potential causal relationships in Mendelian randomization (MR), and (**B**) 314 genes annotated from IVs for pairs of microbial features and H1N1 severity with potential causal relationships in MR. BP: biological process, CC: cellular component, MF: molecular function.

**Table 1 microorganisms-12-01170-t001:** Sensitivity analysis of the causal effect between gut microbiota and H7N9 susceptibility.

Exposure	Method	Heterogeneity			Pleiotropy Egger Intercept Pval	MR–PRESSO Global Test Pval
		Q	Q_df	Q_pval		
*f_Coriobacteriaceae*	MR–Egger	9.371	12	0.671	0.830	0.754
	IVW	9.419	13	0.741		
*f_Leuconostocaceae*	MR–Egger	8.736	9	0.462	1.000	0.592
	IVW	8.736	10	0.557		
*g_Mobiluncus*	MR–Egger	28.293	17	0.042	0.961	0.083
	IVW	28.298	18	0.058		
*g_Rahnella*	MR–Egger	6.428	10	0.778	0.098	0.600
	IVW	9.751	11	0.553		
*s_Citrobacter youngae*	MR–Egger	6.861	10	0.738	0.433	0.766
	IVW	7.529	11	0.755		
*s_Clostridium hylemonae*	MR–Egger	3.534	9	0.939	0.198	0.858
	IVW	5.469	10	0.858		
*s_Clostridium ramosum*	MR–Egger	6.906	13	0.907	0.978	0.930
	IVW	6.907	14	0.938		
*s_Coprococcus catus*	MR–Egger	13.319	13	0.424	0.106	0.347
	IVW	16.410	14	0.289		
*s_Desulfovibrio alaskensis*	MR–Egger	3.362	6	0.762	0.970	0.878
	IVW	3.364	7	0.849		
*s_Eubacterium cellulosolvens*	MR–Egger	0.018	2	0.991	0.773	0.983
	IVW	0.127	3	0.988		
*s_Slackia exigua*	MR–Egger	5.910	12	0.921	0.390	0.913
	IVW	6.706	13	0.917		
*s_Streptococcus mitis*	MR–Egger	7.813	12	0.800	0.184	0.701
	IVW	9.798	13	0.710		
*s_Streptococcus peroris*	MR–Egger	5.576	6	0.472	0.213	0.430
	IVW	7.515	7	0.377		
*s_Streptococcus sanguinis*	MR–Egger	4.667	9	0.862	0.433	0.862
	IVW	5.339	10	0.867		

Abbreviations: IVW, inverse variance weighted; MR–PRESSO, MR–pleiotropy residual sum and outlier. “s_”, “g_”, “f_” are species, genus, family, order, class, and phylum, respectively.

**Table 2 microorganisms-12-01170-t002:** Sensitivity analysis of the causal effect between gut microbiota and H1N1 severity.

Exposure	Method	Heterogeneity			Pleiotropy Egger Intercept Pval	MR–PRESSO Global Test Pval
		Q	Q_df	Q_pval		
*f_Sutterellaceae*	MR–Egger	10.309	17	0.890	0.612	0.911
	IVW	10.576	18	0.911		
*g_Acetivibrio*	MR–Egger	8.999	11	0.622	0.894	0.725
	IVW	9.018	12	0.701		
*g_Ethanoligenens*	MR–Egger	4.722	9	0.858	0.593	0.894
	IVW	5.029	10	0.889		
*g_Subdoligranulum*	MR–Egger	4.104	13	0.990	0.611	0.992
	IVW	4.375	14	0.993		
*g_Yersinia*	MR–Egger	2.935	6	0.817	0.499	0.850
	IVW	3.453	7	0.840		
*s_Bacteroides_thetaiotaomicron*	MR–Egger	15.037	18	0.659	0.366	0.673
	IVW	15.897	19	0.664		
*s_Bacteroides_xylanisolvens*	MR–Egger	6.308	8	0.613	0.408	0.645
	IVW	7.071	9	0.630		
*s_Clostridiales_genomosp._BVAB3*	MR–Egger	9.242	9	0.415	0.719	0.535
	IVW	9.384	10	0.496		
*s_Eubacterium_biforme*	MR–Egger	17.371	22	0.743	0.680	0.790
	IVW	17.546	23	0.782		
*s_Faecalibacterium_prausnitzii*	MR–Egger	24.531	29	0.702	0.203	0.706
	IVW	26.226	30	0.664		
*s_Neisseria_gonorrhoeae*	MR–Egger	5.220	8	0.734	0.793	0.821
	IVW	5.294	9	0.808		
*s_Roseburia_intestinalis*	MR–Egger	1.476	5	0.916	0.331	0.873
	IVW	2.636	6	0.853		
*s_Streptococcus_parasanguinis*	MR–Egger	12.312	18	0.831	0.528	0.862
	IVW	12.726	19	0.852		
*s_Streptococcus_suis*	MR–Egger	7.267	13	0.888	0.754	0.914
	IVW	7.369	14	0.920		
*s_Treponema_vince*	MR–Egger	14.963	15	0.454	0.147	0.413
	IVW	17.306	16	0.366		
*s_Turicibacter_sanguinis*	MR–Egger	3.302	5	0.653	0.382	0.671
	IVW	4.219	6	0.647		
*s_Veillonella_atypica*	MR–Egger	9.639	12	0.648	0.217	0.584
	IVW	11.338	13	0.583		
MF0032:glutamate_degradation_III	MR–Egger	8.588	17	0.952	0.294	0.948
	IVW	9.761	18	0.939		

Abbreviations: IVW, inverse variance weighted; MR–PRESSO, MR–pleiotropy residual sum and outlier. “s_”, “g_”, “f_” are species, genus, family, order, class, and phylum, respectively.

## Data Availability

The raw data supporting the conclusions of this article will be made available by the authors on request.

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
