# Peer review of "Identification of Causal Relationships between Gut Microbiota and Influenza a Virus Infection in Chinese by Mendelian Randomization"

_microorganisms, 2024, doi:10.3390/microorganisms12061170_

Round 1
Reviewer 1 Report
Comments and Suggestions for Authors
Sincere thanks to the authors for their efforts in the article. However, the authors should pay attention to several issues:
1. Keywords should be written alphabetically and in lower case.
2. The bibliography must be adapted to the requirements of the journal.
3. After referring to figure 1, it is a good idea to present it right away.
4. And similarly other figures and tables - it is definitely better readability then, you can immediately look at the graphics/tables when you see them right after the reference in the text.
5. In the table, the names of microorganisms should be in italics.
Comments on the Quality of English Language
Minor correction of English.
Author Response
Sincere thanks to the authors for their efforts in the article. However, the authors should pay attention to several issues:
Response: Thank you for your valuable suggestions on our manuscript. We have carefully addressed each of your points and made the following revisions:
- Keywords should be written alphabetically and in lower case.
Response: Thank you for your comments. Keywords have been arranged alphabetically and converted to lowercase according to your suggestion.
- The bibliography must be adapted to the requirements of the journal.
Response: Thank you for your comments. The bibliography has been adjusted to comply with the journal's requirements.
- After referring to figure 1, it is a good idea to present it right away.
Response: Thank you for your comments. In response to your suggestion, after referring to Figure 1, we have promptly presented it in the text.
- And similarly other figures and tables - it is definitely better readability then, you can immediately look at the graphics/tables when you see them right after the reference in the text.
Response: Thank you for your comments. Likewise, other figures and tables have been positioned immediately after their first quoted paragraph in the main text to enhance readability.
5.In the table, the names of microorganisms should be in italics.
Response: Thank you for your comments. Microorganism names in the table have been italicized as recommended.
We believe that these modifications have significantly improved the clarity and organization of the manuscript. We hope that you find these changes satisfactory.

Reviewer 2 Report
Comments and Suggestions for Authors
The authors’ study indicated the potential existence of a causal relationship between gut microbial features and IAV infection and disease severity. This association may be mediated through regulating cell adherens junctions and signal transduction pathways.
Recent advances in microbiota explorations have led to an improved knowledge of the communities of commensal microorganisms within the human body. Among the relevant inter-organ connections, the gut–lung axis (GLA) remains less studied than the gut–brain axis. So far, microbiota studies mainly focused on the bacterial component.
Q1: A study in patients infected with H7N9 virus showed a reduction in phyla Bacteroidetes and genus including Bacteroides, Blautia, Roseburia, and Ruminococcus but enrichment of Firmicutes and Proteobacteria and genera, including Escherichia, Clostridium, and Enterococcus faecium. Another study, performed in influenza subtype H1N1 patients, reported a depletion of phyla Actinobacteria and Firmicutes, and genera including Dorea, Faecalibacterium, Ruminococcus, Streptococcus together with an enrichment for both Actinomycetaceae and Micrococcaceae. /10.3390/ijms25074051 In a recent review, 11 different studies reported gut microbiome modifications in patients with a proven or suspected respiratory tract infection (RTI), compared to healthy controls. In summary, gut microbiome alterations in patients were consistently in diversity with a depletion of Firmicutes, Lachnospiraceae, Ruminococcaceae and enrichment of Enterococcus 10.1371/journal.pone.0262057 . The authors should add references
Q2: SCFAs can promote gut homeostasis through a positive feedback loop by directing the metabolism of colonic epithelial cells toward fatty acid ß-oxydation. Under homeostatic conditions, microbiota-derived butyrate is sensed by the nuclear protein peroxisome proliferator-activated receptor gamma (PPARγ) in colonic epithelial cells. 10.1128/MCB.00858-12. Signaling through this transcription factor directs the energy metabolism of colonocytes to FAO and oxidative phosphorylation (OXPHOS) in mitochondria10.1016/j.bbadis.2011.02.014. These two metabolic pathways require high oxygen consumption resulting in epithelial hypoxia 10.1126/science.aam9949. The authors should add discussions and references regarding these mechanisms.
Q3: SCFAs can also promote T cell differentiation into T helper (Th) 1 and Th17 effector cells and IL-10+Tregs, independent of GPR41 and GPR43. In this case, SCFAs act as HDAC inhibitors to enhance the mTOR–S6K pathway required for T cell differentiation and cytokine expression 10.1038/mi.2014.44. The authors should add references
Comments on the Quality of English LanguageMinor English Editing is required.
Author Response
The authors’ study indicated the potential existence of a causal relationship between gut microbial features and IAV infection and disease severity. This association may be mediated through regulating cell adherens junctions and signal transduction pathways.
Recent advances in microbiota explorations have led to an improved knowledge of the communities of commensal microorganisms within the human body. Among the relevant inter-organ connections, the gut–lung axis (GLA) remains less studied than the gut–brain axis. So far, microbiota studies mainly focused on the bacterial component.
Response: Thank you for your thorough review and insightful comments on our manuscript. We have thoroughly considered the references you suggested to add and have made the following modifications to the article:
Q1: A study in patients infected with H7N9 virus showed a reduction in phyla Bacteroidetes and genus including Bacteroides, Blautia, Roseburia, and Ruminococcus but enrichment of Firmicutes and Proteobacteria and genera, including Escherichia, Clostridium, and Enterococcus faecium. Another study, performed in influenza subtype H1N1 patients, reported a depletion of phyla Actinobacteria and Firmicutes, and genera including Dorea, Faecalibacterium, Ruminococcus, Streptococcus together with an enrichment for both Actinomycetaceae and Micrococcaceae. /10.3390/ijms25074051 In a recent review, 11 different studies reported gut microbiome modifications in patients with a proven or suspected respiratory tract infection (RTI), compared to healthy controls. In summary, gut microbiome alterations in patients were consistently in diversity with a depletion of Firmicutes, Lachnospiraceae, Ruminococcaceae and enrichment of Enterococcus 10.1371/journal.pone.0262057 . The authors should add references
Response: Thank you for your comments. In the Introduction and Discussion sections, we have added references ([13][18]).
“A systematic review of gut microbiota changes in respiratory tract infection (RTI) patients consistently revealed decreased diversity, with depletion of Firmicutes, Lachnospiraceae, Ruminococcaceae, and Ruminococcus, and enrichment of Enterococcus [13].” (Line 64-66)
“Investigations have revealed significant changes in the composition of intestinal microbiota in individuals affected by different IAV subtypes, including H7N9 and H1N1 [16-18].” ( Line 70-72)
“Previous studies have reported a reduction in Firmicutes among patients with RTIs, including H1N1 infection, COVID-19, tuberculosis (TB), community-acquired pneumonia, and recurrent respiratory tract infections [13].” ( Line 365-367)
Q2: SCFAs can promote gut homeostasis through a positive feedback loop by directing the metabolism of colonic epithelial cells toward fatty acid ß-oxydation. Under homeostatic conditions, microbiota-derived butyrate is sensed by the nuclear protein peroxisome proliferator-activated receptor gamma (PPARγ) in colonic epithelial cells. 10.1128/MCB.00858-12. Signaling through this transcription factor directs the energy metabolism of colonocytes to FAO and oxidative phosphorylation (OXPHOS) in mitochondria10.1016/j.bbadis.2011.02.014. These two metabolic pathways require high oxygen consumption resulting in epithelial hypoxia 10.1126/science.aam9949. The authors should add discussions and references regarding these mechanisms.
Response: Thank you for your comments. We have added references ([58-60]) in the Discussion section regarding the mechanism by which SCFAs direct the energy metabolism of colonic epithelial cells towards fatty acid oxidation and promote intestinal homeostasis.
“Microbiota-derived butyrate activates the nuclear receptor peroxisome proliferator activated receptor γ in colonic epithelial cells, shifting their energy metabolism towards fatty acid oxidation and oxidative phosphorylation in mitochondria, which consumes high levels of oxygen [58-60]. This suggests that SCFAs promotes gut homeostasis through a positive feedback loop by limiting the luminal bioavailability of oxygen.” ( Line 371-375)
Q3: SCFAs can also promote T cell differentiation into T helper (Th) 1 and Th17 effector cells and IL-10+Tregs, independent of GPR41 and GPR43. In this case, SCFAs act as HDAC inhibitors to enhance the mTOR–S6K pathway required for T cell differentiation and cytokine expression 10.1038/mi.2014.44. The authors should add references.
Response: Thank you for your comments. We have also added the referenced article ([63]) to support the statement regarding SCFAs promoting T cell differentiation and acting as HDAC inhibitors in the Discussion section.
“Moreover, SCFAs can induce both effector and regulatory T cells to regulate the immune system by inhibiting the direct histone deacetylase (HDAC) in T cells and enhancing the mTOR-S6K pathway [63].” ( Line 379-382)
We believe that these modifications have significantly improved the clarity and organization of the manuscript. We hope that you find these changes satisfactory.

Reviewer 3 Report
Comments and Suggestions for Authors
The manuscript provides valuable insights into the causal relationship between gut microbiota and influenza A virus (IAV) infection. The study uses a robust methodology, Mendelian randomization (MR), to answer this important question and identify specific bacterial taxa associated with susceptibility to H7N9 and severity of H1N1 infection. The results have significant implications for understanding the pathogenesis of IAV infection and the potential to modulate the gut microbiota to prevent or ameliorate IAV disease. The text has some weaknesses that the authors should pay attention to when discussing the results. The sample size for the H1N1 and H7N9 GWAS is relatively small, which may limit the generalizability of the results. The study population is exclusively of Chinese descent, which may limit the applicability of the results to other ethnic groups. The authors could further strengthen the manuscript by discussing the possible mechanisms by which the identified bacterial taxa influence IAV infection risk and outcomes. The authors could also discuss the limitations of their study in more detail and suggest directions for future research.
· Introduction – The Chinese population is mentioned in the title, but the focus is not explicitly stated in the introduction. Mention why the study is specific to the Chinese population. Consider mentioning specific components of the gut microbiota that may be involved in IAV infection. Explain how MR helps eliminate confounding factors in the relationship between gut microbiota and IAV. The last sentence about mechanisms is vague. Mention what biological mechanisms could be explored.
· Materials and Methods –
o Section 2.1. – The explanation of Mendelian randomization (MR) analysis should be provided in more detail in terms of the specific methods and rationale for the statistical tests chosen. Improve the flow and clarity of explaining the three instrumental variable (IVs) assumptions.
o Section 2.3. – Provide more detailed information on patient selection criteria and diagnostic protocols.
o Section 2.5. – Provide more details about IVW test, MR-Egger regression and weighted median method.
o Section 2.7. – Clarify the use of PheWAS and its relevance to the study results. Provide more detailed information about the enrichment analysis methods and their importance.
· Results – Provide more context or explanations to key findings, particularly when introducing specific bacterial taxa and their associations. Combine and rationalize the sentences in lines 193-197 for clarity. Similarly, combine related results in lines 201-204 for better flow. Briefly summarize the key findings from the PPI network and PheWAS results in lines 270-280. Briefly summarize the key findings from the PPI network and PheWAS results in lines 270-280.
· Discussion – Explain how specific bacterial taxa influence the immune response and what implications this has for therapy. Provide more context for the genetic analyzes and their biological significance. Strengthen discussion about how these findings might inform future research and clinical applications.
· Conclusions – The first sentence could address the identified causal relationships more specifically. Consider adding specific examples of the identified bacterial characteristics associated with IAV. Mention the possible mechanisms involving adherent connections and signaling pathways discussed in the Discussion section. Strengthen the connection between the conclusions and the title by mentioning the use of Mendelian randomization to establish causality. Briefly note the limitations mentioned in the Discussion section. You may consider adding a sentence or two at the end to mention the potential implications for future research or therapeutic development.
· Abstract – Consider adding a sentence about the study population. Consider reinforcing the novelty aspect. You can shorten the sentence about Clostridium hylemonae and Faecalibacterium prausnitzii a little. You may consider adding a sentence or two at the end to mention the potential implications for future research or therapeutic development.
· In general – Please ensure that all names of microorganisms in the manuscript are written in italics where necessary.
Comments on the Quality of English Language
Minor editing of English language required.
Author Response
The manuscript provides valuable insights into the causal relationship between gut microbiota and influenza A virus (IAV) infection. The study uses a robust methodology, Mendelian randomization (MR), to answer this important question and identify specific bacterial taxa associated with susceptibility to H7N9 and severity of H1N1 infection. The results have significant implications for understanding the pathogenesis of IAV infection and the potential to modulate the gut microbiota to prevent or ameliorate IAV disease. The text has some weaknesses that the authors should pay attention to when discussing the results. The sample size for the H1N1 and H7N9 GWAS is relatively small, which may limit the generalizability of the results. The study population is exclusively of Chinese descent, which may limit the applicability of the results to other ethnic groups. The authors could further strengthen the manuscript by discussing the possible mechanisms by which the identified bacterial taxa influence IAV infection risk and outcomes. The authors could also discuss the limitations of their study in more detail and suggest directions for future research.
Response: We are grateful to the reviewer for pointing out these issues. According to your suggestions, we have made some modifications to each section of the manuscript, as detailed below.
- Introduction – The Chinese population is mentioned in the title, but the focus is not explicitly stated in the introduction. Mention why the study is specific to the Chinese population. Consider mentioning specific components of the gut microbiota that may be involved in IAV infection. Explain how MR helps eliminate confounding factors in the relationship between gut microbiota and IAV. The last sentence about mechanisms is vague. Mention what biological mechanisms could be explored.
Response: Thank you for your comments. We emphasized that this study is specific to the Chinese population (line 87-89). Additionally, we have included mention of specific components of the gut microbiota that may be involved in IAV infection to provide further clarity on the research focus (line 63-65). We have also provided an explanation of how MR helps eliminate confounding factors in the relationship between gut microbiota and IAV (line 77-82). Lastly, we have deleted the sentence aboutbiological mechanisms.
“A systematic review of gut microbiota changes in respiratory tract infection (RTI) patients consistently revealed decreased diversity, with depletion of Firmicutes, Lachnospiraceae, Ruminococcaceae, and Ruminococcus, and enrichment of Enterococcus [13].” ( Line 63-65)
“Genetic variants are randomly assigned during meiosis and not influenced by traditional confounding factors such as environment, socioeconomic status, and behavior. Furthermore, genetic variants remain stable after birth, enabling the evaluation of chronologically plausible associations. Consequently, MR can overcome confounding and reverse causality issues inherent in traditional observational studies.” ( Line 77-82)
“Xu et al. employed MR to investigate the causal effect of gut microbiota on seasonal influenza and influenza-induced pneumonia in Finnish. In their study, the identification of intestinal flora was limited to the genus level [22]. In this study, we performed a systematic two-sample MR study to explore the causal effect of species-level gut microbiota on avian IAV H7N9 susceptibility and H1N1pdm09 severity in Chinese. Note that genome-wide association studies (GWAS) investigating H7N9 susceptibility have been limited to Chinese populations thus far. The outcomes of this study might offer new insights for personalized IAV treatment through the regulation of gut microbiota.” ( Line 84-92)
- Materials and Methods –
o Section 2.1. – The explanation of Mendelian randomization (MR) analysis should be provided in more detail in terms of the specific methods and rationale for the statistical tests chosen. Improve the flow and clarity of explaining the three instrumental variable (IVs) assumptions.
o Section 2.3. – Provide more detailed information on patient selection criteria and diagnostic protocols.
o Section 2.5. – Provide more details about IVW test, MR-Egger regression and weighted median method.
- Section 2.7. – Clarify the use of PheWAS and its relevance to the study results. Provide more detailed information about the enrichment analysis methods and their importance.
Response: Thank you for your comments. We have revised and expanded these sections to provide more detailed explanations of Mendelian randomization analysis, patient selection criteria, statistical methods (IVW, MR-Egger, WM), PheWAS and enrichment analysis. These enhancements aim to improve reader understanding of our methodology.
o Section 2.1. – “First, we conducted a two-sample MR analysis to investigate potential causal relationships between gut microbial features and IAV infection. Three different MR methods (refer to Methods further below) were performed to increase the robustness of the results and avoid bias. The selection of IVs adhered to three critical assumptions [19]: (i) Relevance: the IVs were associated with the exposure of interest; (ii) Independence: the IVs were independent with other confounding factors that affect both the exposure and outcome; (iii) Exclusion restriction: the IVs were required to influence the outcome solely through the studied exposure. The stringent criteria for IV selection were essential to ensure the validity and robustness of the MR analysis results.” ( Line 95-103)
o Section 2.3. – “Real-time reverse transcription polymerase chain reaction (RT-PCR) was used for diagnosing H7N9 infection, following the Diagnostic and Treatment Protocol for Human Infection with Avian Influenza A (H7N9) (http://www.nhc.gov.cn/gjhzs/s7952/201304/34750c7e6930463aac789b3e2156632f.shtml).” ( Line 124-127)
o Section 2.5. – “Suppose there are M genetic variants, where j represents the jth variant, x represents the exposure, and y represents the outcome. The IVW method is a basic model assuming that pleiotropy does not exist or is zero. It estimated the causal effect () by integrating the effect ratio () of genetic variant on outcome () and exposure () using inverse variance weighting. IVW estimates can also obtained using the following weighted linear regression model without intercept terms:
|
, |
(3) |
The estimate of the slope parameter b is the causal effect, and the weight is the inverse of the variance of the genetic association effect corresponding to the outcome. In order to control the uncorrelated pleiotropy, MR-Egger identifies the violations of pleiotropy and heterogeneity by incorporating an intercept in the IVW model [30]. The regression model is as follows:
|
, |
(4) |
Because some IVs may be pleiotropic, the WM method estimates the causal effect from the weighted median of the effect ratio ().” ( Line 158-171)
o Section 2.7. –“PheWAS offered a more comprehensive understanding of biological significance of these genes and helped elucidate the MR results.” ( Line 197-198)
“Additionally, Gene Ontology (GO) [42,43] and Kyoto Encyclopedia of Genes and Genomes (KEGG) [44] pathway enrichment analyses were performed for the positional mapped genes using DAVID (https://david.ncifcrf.gov/tools.jsp) [45,46] to reveal the potential underlying biological pathways or functions of casual associations. GO is utilized as a bioinformatics tool for annotating genes and analyzing the biological processes they are involved in. It classifies gene functions into biological processes (BP), molecular functions (MF), and cellular components (CC). KEGG is a database for analyzing molecular signaling pathways and interactions in biological systems.” ( Line 200-208)
“The enrichment analysis results are visualized by SRplot web server (http://www.bioinformatics.com.cn/SRplot) [47].” ( Line 209-210)
- Results – Provide more context or explanations to key findings, particularly when introducing specific bacterial taxa and their associations. Combine and rationalize the sentences in lines 193-197 for clarity. Similarly, combine related results in lines 201-204 for better flow. Briefly summarize the key findings from the PPI network and PheWAS results in lines 270-280.
Response: Thank you for your comments. We have provided additional context and explanations for key findings, especially regarding specific bacterial taxa and their associations. Additionally, we have combined and rationalized sentences in Result section to enhance clarity and flow. We have also summarized the key findings from the PPI network and PheWAS results concisely.
“It is worth noting that Clostridium hylemonae [β = -0.335, 95% CI: -0.622 to -0.049, p = 0.022] and Clostridium ramosum [β = -0.390, 95% CI: -0.684 to -0.096, p = 0.009], known for synthesizing short-chain fatty acids, exhibit a negative correlation with the susceptibility to H7N9. Additionally, Streptococcus peroris [β = -0.327, 95% CI: -0.648 to -0.006, p = 0.046] and Streptococcus sanguinis [β = -0.552, 95% CI: -1.017 to -0.087, p = 0.020]) were associated with a reduced risk of H7N9 susceptibility, while Streptococcus mitis (β = 0.504, 95% CI: 0.022 to 0.986, p = 0.041) showed an increased risk of H7N9 susceptibility.” (Line 222-228)
“In particular, Faecalibacterium prausnitzii [β = -0.394, 95% CI: -0.774 to -0.013, p = 0.043], playing a key role in the biosynthesis of short-chain fatty acids, exhibit a negative association with the risk of H1N1 severity. Moreover, Streptococcus parasanguinis [β = -0.574, 95% CI: -1.004 to -0.144, p = 0.009] and Streptococcus suis [β = -0.455, 95% CI: -0.865 to -0.044, p = 0.030] were also correlated with a reduced risk of H1N1 severity.” (Line 252-256)
“In summary, PPI networks and PheWAS analysis uncover the connection between IAV infection and microbial features, which are associated with hub genes as well as the metabolic or nutritional domain and the immunological domain.” ( Line 297-299)
- Discussion – Explain how specific bacterial taxa influence the immune response and what implications this has for therapy. Provide more context for the genetic analyzes and their biological significance. Strengthen discussion about how these findings might inform future research and clinical applications.
Response: Thank you for your comments. We have addressed your feedback by elucidating the influence of specific bacterial taxa, providing detailed explanations for genetic analyses and their biological significance, and enhancing the discussion on how these findings can guide future research and clinical applications.
“Microbiota-derived butyrate activates the nuclear receptor peroxisome proliferator activated receptor γ in colonic epithelial cells, shifting their energy metabolism towards fatty acid oxidation and oxidative phosphorylation in mitochondria, which consumes high levels of oxygen [58-60]. This suggests that SCFAs promotes gut homeostasis through a positive feedback loop by limiting the luminal bioavailability of oxygen.” ( Line 371-375)
“Moreover, SCFAs can induce both effector and regulatory T cells to regulate the immune system by inhibiting the direct histone deacetylase (HDAC) in T cells and enhancing the mTOR-S6K pathway [63].” (Line 379-382)
“The LDL receptor family has roles related to the clearance of extracellular ligands and is proposed to be involved in extracellular signal transduction [74].” (Line 419-421)
“PTPs are known to be signaling molecules that regulate a variety of cellular processes, including cell growth, differentiation, mitotic cycle, and oncogenic transformation [78].” ( Line 425-426)
“Thirdly, our findings were not corroborated by in vivo flora colonization assay.” (Page 15, Line 465-466)
“Despite these limitations, we believe that our findings provided new clues for further investigation of microbial function via microbiota colonization in vivo. Focusing on gut microbiota could represent an innovative approach to the prophylaxis and therapy of IAV infections.” (Page 15, Line 467-470)
- Conclusions – The first sentence could address the identified causal relationships more specifically. Consider adding specific examples of the identified bacterial characteristics associated with IAV. Mention the possible mechanisms involving adherent connections and signaling pathways discussed in the Discussion section. Strengthen the connection between the conclusions and the title by mentioning the use of Mendelian randomization to establish causality. Briefly note the limitations mentioned in the Discussion section. You may consider adding a sentence or two at the end to mention the potential implications for future research or therapeutic development.
Response: Thank you for your comments. According to your advice, we have revised the conclusion section entirely, as follows:
“Our MR study has successfully identified the potential causal effects of 12 and 15 gut microbial features on H7N9 infection and H1N1pdm09 severity, respectively. In particular, Clostridium hylemonae and Faecalibacterium prausnitzii, which promote the production of SCFAs, were negatively associated with H7N9 susceptibility and H1N1pdm09 severity separately. These findings not only highlight the significant implications for understanding the pathogenesis of IAV infection but also provides valuable clues for future research to elucidate the role of gut microbiota in infectious diseases.
In summary, our study not only expands our knowledge concerning the impact of the gut microbiota on H7N9 susceptibility and H1N1pdm09 severity but also underscores the potential for targeting gut microbial composition based on these findings. The causally related bacterial groups open a promising avenue for the development of novel preventative and therapeutic strategies against IAV and other respiratory tract infections. An in-depth comprehension of the foundational mechanisms will facilitate the development of efficacious strategies to modulate the gut microbiome, thereby reducing the incidence and severity of IAV infections.” (Page 15, Line 472-486)
- Abstract – Consider adding a sentence about the study population. Consider reinforcing the novelty aspect. You can shorten the sentence about Clostridium hylemonae and Faecalibacterium prausnitzii a little. You may consider adding a sentence or two at the end to mention the potential implications for future research or therapeutic development.
Response: Thank you for your comments. We have emphasized that this study has focused on the Chinese population, streamlined the description of the results, and elucidated the significance of the study and future prospects at the end .
“This investigation aimed to assess the influence of gut microbiota on susceptibility to human infection with H7N9 avian IAV and the severity of influenza A (H1N1)pdm09 infection. A two-sample Mendelian randomization analysis was conducted, integrating our in-house genome-wide association study (GWAS) on H7N9 susceptibility and H1N1pdm09 severity with a metagenomics GWAS dataset from Chinese population. Twelve and fifteen gut microbiotas causally associated with H7N9 susceptibility or H1N1pdm09 severity, separately. Notably, Clostridium hylemonae and Faecalibacterium prausnitzii were negative associated with H7N9 susceptibility and H1N1pdm09 severity, respectively. Moreover, Streptococcus peroris and Streptococcus sanguinis were associated with H7N9 susceptibility, while Streptococcus parasanguini and Streptococcus suis were correlated with H1N1pdm09 severity. These results provide novel insights into the interplay between gut microbiota and IAV pathogenesis as well as new clues for mechanism research regarding therapeutic interventions or IAV infections. Future studies should concentrate on clarifying the regulatory mechanisms of gut microbiota and developing efficacious approaches to reduce the incidence of IAV infectious, which could improve strategy for preventing and treating IAV infection worldwide.” (Line 22-36)
- In general – Please ensure that all names of microorganisms in the manuscript are written in italics where necessary.
Response: Thank you for your comments. We have checked the names of microorganisms throughout the manuscript and modified them to be italicized.
We believe that these modifications have significantly improved the clarity and organization of the manuscript. We hope that you find these changes satisfactory.
